# Interaction of S100A6 Protein with the Four-Helical Cytokines

**DOI:** 10.3390/biom13091345

**Published:** 2023-09-04

**Authors:** Alexey S. Kazakov, Evgenia I. Deryusheva, Victoria A. Rastrygina, Andrey S. Sokolov, Maria E. Permyakova, Ekaterina A. Litus, Vladimir N. Uversky, Eugene A. Permyakov, Sergei E. Permyakov

**Affiliations:** 1Pushchino Scientific Center for Biological Research of the Russian Academy of Sciences, Institute for Biological Instrumentation, Institutskaya str., 7, Pushchino, Moscow Region 142290, Russia; fenixfly@yandex.ru (A.S.K.); janed1986@ya.ru (E.I.D.); certusfides@gmail.com (V.A.R.); 212sok@gmail.com (A.S.S.); mperm1977@gmail.com (M.E.P.); ealitus@gmail.com (E.A.L.); epermyak@yandex.ru (E.A.P.); 2Department of Molecular, Morsani College of Medicine, University of South Florida, Tampa, FL 33612, USA; 3USF Health Byrd Alzheimer’s Research Institute, Morsani College of Medicine, University of South Florida, Tampa, FL 33612, USA

**Keywords:** cytokine, EF hand, S100 protein, S100A6, protein–protein interaction

## Abstract

S100 is a family of over 20 structurally homologous, but functionally diverse regulatory (calcium/zinc)-binding proteins of vertebrates. The involvement of S100 proteins in numerous vital (patho)physiological processes is mediated by their interaction with various (intra/extra)cellular protein partners, including cell surface receptors. Furthermore, recent studies have revealed the ability of specific S100 proteins to modulate cell signaling via direct interaction with cytokines. Previously, we revealed the binding of ca. 71% of the four-helical cytokines via the S100P protein, due to the presence in its molecule of a cytokine-binding site overlapping with the binding site for the S100P receptor. Here, we show that another S100 protein, S100A6 (that has a pairwise sequence identity with S100P of 35%), specifically binds numerous four-helical cytokines. We have studied the affinity of the recombinant forms of 35 human four-helical cytokines from all structural families of this fold to Ca^2+^-loaded recombinant human S100A6, using surface plasmon resonance spectroscopy. S100A6 recognizes 26 of the cytokines from all families of this fold, with equilibrium dissociation constants from 0.3 nM to 12 µM. Overall, S100A6 interacts with ca. 73% of the four-helical cytokines studied to date, with a selectivity equivalent to that for the S100P protein, with the differences limited to the binding of interleukin-2 and oncostatin M. The molecular docking study evidences the presence in the S100A6 molecule of a cytokine-binding site, analogous to that found in S100P. The findings argue the presence in some of the promiscuous members of the S100 family of a site specific to a wide range of four-helical cytokines. This unique feature of the S100 proteins potentially allows them to modulate the activity of the numerous four-helical cytokines in the disorders accompanied by an excessive release of the cytokines.

## 1. Introduction

S100 is an evolutionary young family of structurally similar, but functionally diversified regulatory Ca^2+^-binding proteins of the EF-hand superfamily (for reviews, see [1,2,3,4]). The classical Ca^2+^-binding motif of the ‘EF-hand’ type (PROSITE [5] entry PDOC00018) is composed of a 12-residue Ca^2+^-coordinating loop located between two α-helices [6,7,8]. S100 proteins consist of a low-affinity non-classical *N*-terminal EF-hand motif, and a high-affinity classical *C*-terminal EF-hand, which are linked by a flexible ‘hinge’ [7,9]. Certain S100 proteins possess distinct Zn^2+^/Cu^2+^/Mn^2+^-binding sites [9,10]. S100 proteins are generally (homo/hetero)dimeric proteins (except for monomeric S100G), whereas some of them tend to form higher-order oligomers [2,11]. With the exception of the S100 fused-type proteins, the human S100 family consists of 21 members (78–113 residues; the pairwise sequence identity evaluated via Clustal Omega 2.1 [12] ranges from 16% to 61%), each of which has a unique set of functional activities, together covering nearly all vital processes [1,2,3]. Some S100 proteins are linked to the progression of oncological, autoimmune, inflammatory, neurodegenerative, cardiovascular, pulmonary, and liver diseases, are used for diagnostic and prognostic purposes, and are considered as promising therapeutic targets [13,14,15,16,17,18,19,20,21]. The dysregulation of most S100 proteins in cancer is related to the fact that most of their genes are located in the epidermal differentiation complex on chromosome 1 (locus 1q21), which undergoes frequent rearrangement in cancer [22,23]. These S100 proteins are designated by Arabic numbers placed behind the ‘S100A’, while the names of the S100 proteins encoded by other chromosomes carry the symbol ‘S100’ followed by ‘B’, ‘G’, ‘P’, or ‘Z’ [23,24]. The multifunctionality of S100 proteins is determined by their cell-/tissue-specific expression, ability to localize in the nucleus/cytosol/extracellular space, metal-binding properties, post-translational modifications, and propensity to recognize a wide range of targets, including enzymes, transcription factors, receptor/membrane proteins, lipids, and nucleic acids [1,2,3,25,26].

Despite the lack of a leader sequence, some of the S100 proteins are secreted via the endoplasmic reticulum (ER)/Golgi pathway, or several unconventional passive and active mechanisms [2,27,28]. When they are released into the extracellular space, certain S100 proteins exert a cytokine-like action, due to interaction with specific cell surface receptors, such as RAGE, TLR4, ErbB1, ErbB3, ErbB4, IL-10R, integrin β1, neuroplastin-β, 5-HT_1B_, 4-HT_4_, SIRL-1, ALCAM, EMMPRIN, CD33, CD36, CD68, CD69, or CD146 [2,25,29,30,31,32,33,34,35,36]. Moreover, some of the released S100 proteins interact with cytokines. For instance, S100A4 binds ErbB1/ErbB4 ligands [35,37], S100A13 interacts with IL1α/FGF1 [38,39], S100B binds FGF2 [40,41], and S100A11/A12/A13 interact with the soluble form of TNF [42]. Several S100 proteins recognize four-helical cytokines: S100A2/A6/P bind EPO [43], S100A1/A4/A6/B/P interact with IFN-β [25,44,45], and distinct subsets of S100A1/A6/B/P bind to specific IL-6 family cytokines (IL-11, OSM, CNTF, CT-1, and CLCF1 [46]), whereas S100P binds 29 four-helical cytokines [47]. The S100 binding has been shown to affect the cytokine signaling in some cases: S100A4 enhanced the amphiregulin-mediated proliferation of embryonic fibroblasts [37]; S100B inhibited an FGF2-induced increase in the proliferation of MCF-7 and MDA-MB468 cells [40], but favored the FGF2-medited activation of FGFR1 in myoblasts [41,48]; S100A12/A13 rescued Huh-7 cells from the cytotoxic effect of soluble TNF [42]; and the S100A1/A4/B/P proteins suppressed the IFN-β-induced inhibition of viability in MCF-7 cells [25,44,45]. Moreover, S100-cytokine interactions can promote the non-classical secretion of the cytokines, as demonstrated by S100A13–IL1α/FGF1 interactions [38,39].

The vast majority of the established S100–cytokine interactions correspond to cytokines belonging to the superfamily of four-helical cytokines (SCOP [49] ID: 3001717), which is subdivided into three families: ‘Short-chain cytokines’ (SCOP ID 4000852), ‘Long-chain cytokines’ (SCOP ID 4000851), and ‘Interferons/interleukin-10 (IL-10)’ (SCOP ID 4000854). We have previously shown that the S100P oncoprotein is poorly selective towards the four-helical cytokines of all structural families, interacting with 71% of them (29 out of the 41 cytokines studied), with the equilibrium dissociation constants, K_d_, in the 1 nM–3 µM range (below the K_d_ value for S100P binding to the V domain of RAGE) [47]. Using mutagenesis, we confirmed the presence of a cytokine-binding site in the S100P molecule, overlapping with its RAGE-specific site. The ability of S100P to recognize multiple cytokines reflects its promiscuous nature, well established for certain members of the S100 family, which readily share their binding partners [50,51]. Therefore, the specificity to many four-helical cytokines is also expected for other promiscuous representatives of the S100 family. To explore this possibility, in this work, we test the affinity of another promiscuous S100 oncoprotein [50,51], S100A6 (also known as calcyclin; it has a pairwise sequence identity to S100P of 35% [52,53]), specific to the six four-helical cytokines shown in Table 1, to 35 four-helical cytokines from all their structural families (Appendix A). Our findings greatly expand the long list of known extracellular target proteins of S100A6, which may be of value for deciphering its role in the progression of many cancers, neurodegenerative disorders, myocardial infarction, acute coronary syndrome, chronic renal disease, primary biliary cholangitis, pulmonary fibrosis, systemic sclerosis of the lung, endometriosis, osteoarthritis, various eye pathologies, and other disorders (reviewed in [54,55,56]).

## 2. Materials and Methods

### 2.1. Materials

The human S100A6 protein was expressed in *E. coli*, and purified as described in [44]. The samples of human cytokines are presented in Appendix A. The protein concentrations were determined according to [57].

The HEPES and sodium chloride were from PanReac AppliChem (Darmstadt, Germany). The CaCl_2_, Tween 20, and EDTA were purchased from Sigma Aldrich Co. (Burlington, MA, USA).

### 2.2. Surface Plasmon Resonance Studies

SPR studies of S100A6 binding to the cytokines immobilized on the sensor chip surface (S100A6 was used as an analyte, and the cytokines served as ligands) were performed at 25 °C, mainly according to [47]. The cytokine (0.03–0.05 mg/ml) was immobilized on the ProteOn™ GLH sensor chip surface of the ProteOn™ XPR36 instrument (Bio-Rad Laboratories, Inc. Hercules, CA, USA) via amine coupling (up to 10,000–17,000 RUs). The running buffer was 10 mM HEPES, 150 mM NaCl, 1 mM CaCl_2_, and 0.05% Tween 20, pH 7.4. The S100A6 (63 nM–8 µM) solution in the running buffer was passed over the chip surface for 300 s. The dissociation of the ligand–analyte complex was triggered by the passage of the running buffer for 1200–2400 s. The SPR sensograms were described via a single-site binding scheme (S100A6-IL-2 interaction) or a heterogeneous ligand model (1). The latter suggests the presence of two populations of the ligand (L_1_ and L_2_) independently binding an analyte molecule (A):(1)$$L_{1}+A\overset{k_{a1}}{\underset{\underset{K_{d1}}{k_{d1}}}{⇄}}L_{1}A\;\;L_{2}+A\overset{k_{a2}}{\underset{\underset{K_{d2}}{k_{d2}}}{⇄}}L_{2}A$$

Here, k and K refer to the kinetic and equilibrium association (‘a’) and dissociation (‘d’) constants, respectively. The K_d_ and k_d_ values were calculated for several S100A6 concentrations. The resulting values are the average of 3–5 estimates. The ligand was regenerated using 20 mM EDTA pH 8.0 solution for 100 s. The free energy change in the reaction was estimated using the equation: ΔG_i_ = −RT ln(55.3/K_di_), i=1,2.

### 2.3. Structural Classification of Cytokines

The cytokines investigated in this study were classified according to the SCOP 2 database (https://scop2.mrc-lmb.cam.ac.uk [49], build 1.0.6, updated on 06-29-2022, accessed on 1 June 2023) as belonging to the ‘All alpha proteins’ structural class, ‘4-helical cytokines’ fold (SCOP ID 2001054), ‘4-helical cytokines’ superfamily (SCOP ID 3001717). This superfamily comprises three families, namely ‘Short-chain cytokines’ (SCOP ID 4000852), ‘Long-chain cytokines’ (SCOP ID 4000851), and ‘Interferons/interleukin-10 (IL-10)’ (SCOP ID 4000854). CLCF1 and CT-1 were classified as long-chain cytokines, according to [58].

### 2.4. Structural Modeling of The S100A6–Cytokine Complexes

The ClusPro docking server [59] was used to generate the models of the tertiary structures of the S100A6–cytokine complexes, according to [43,47]. The tertiary structure of the Ca^2+^-bound S100A6 dimer was taken from PDB [60] entry 1K9K (X-ray, chains A and B [61]), while the structures of the cytokines were either extracted from PDB, or predicted via AlphaFold2 (https://alphafold.ebi.ac.uk/; accessed on 1 June 2023 [62]) (Appendix A). Ten docking models were generated for each S100A6–cytokine complex. The contact residues included in at least five models were considered as the residues of the binding site. The distributions of the contact residues of S100A6 over its amino acid sequence within the models of the complexes were calculated as described in [43]. The models were visualized using PyMOL v.2.5.0 (https://pymol.org/2/; accessed on 1 June 2023) software.

## 3. Results and Discussion

### 3.1. Selectivity of S100A6 Binding to The Four-Helical Cytokines

We have studied the affinity of the Ca^2+^-loaded (1 mM CaCl_2_) recombinant human S100A6 protein to the panel of 35 recombinant human four-helical cytokines (including the panel used in [47], extended with Flt3L, SCF, and IL-19), which covers all structural families in this fold (Appendix A). The cytokines were immobilized on the surface of the SPR sensor chip via amine coupling, and 61 nM–8 µM solutions of S100A6 were passed over the surface at 25 °C. Whereas nine cytokines (Appendix A) did not reveal noticeable effects in response to the S100A6, the SPR sensograms for 26 cytokines showed the S100A6 concentration-dependent effects, characteristic of association–dissociation processes (Figure 1, Figure 2, Figure 3 and Figure 4). The kinetic SPR data were adequately described within the heterogeneous ligand model (1) (Figure 1, Figure 2, Figure 3 and Figure 4, Table 2), which was earlier successfully used for the description of the S100–cytokine interactions [25,43,44,45,46,47,63,64]. The data for IL-2 were described via the one-site binding scheme. The respective equilibrium dissociation constants, K_d_, range from 0.3 nM (in the case of THPO, similarly to S100P [47]) to 12 µM (for IL-2) (Table 2). For comparison, the reported SPR estimates of the K_d_ values for the complexes of Ca^2+^-bound S100A6 with various extracellular fragments of its receptor, RAGE, are within the range from 28 nM to 13 µM [65,66], whereas the K_d_ value for the V domain of RAGE measured via isothermal calorimetry is 3 µM [67]. An analysis of the free energy changes upon the S100A6–cytokine interactions (Figure 5), ΔG, shows that the average S100A6 affinities for the cytokines of different SCOP families in the fold decrease in the following order: short-chain cytokines > long-chain cytokines > interferons/IL-10 (ΔG of −49.8 kJ/mol < −47.1 kJ/mol < −46.8 kJ/mol). The same tendency was observed previously for the S100P protein [47].

The S100A6–cytokine complexes easily dissociated upon the removal of Ca^2+^ via the passage over the SPR chip of 20 mM EDTA solution, pH 8.0, which evidences the importance of the Ca^2+^-induced structural rearrangement within the S100A6 molecule for efficient interaction with the cytokines. As Ca^2+^ binding induces the solvent exposure of S100A6 residues of helices α2 and α3 and the ‘hinge’ between them [61], this region is likely involved in the cytokine recognition.

As the equilibrium homodimer dissociation constant for the Ca^2+^-loaded S100A6 does not exceed 0.5 µM [44], our estimates of its affinities to the cytokines (Table 2) mostly correspond to the dimeric state of S100A6. Meanwhile, serum S100A6 concentrations may be as low as 0.2 pM [68] (Appendix A), which indicates that, in serum, S100A6 may exist in a monomeric form. 

As the affinities of the S100P monomer to the four-helical cytokines IL-11 and IFN-β exceed those of the S100P dimer by 1.4–2.2 orders of magnitude [25,64,69], S100A6 monomerization is likely to improve its affinity to the four-helical cytokines. In this case, the K_d_ values for some of the S100A6–cytokine interactions (Flt3L, IL-3, IL-5, IL-9, IL-13, IL-15, IL-21, SCF, THPO; G-CSF, GH, GH-V, IL-31, PRL; IL-19, IL-20, IL-26) may reach the nanomolar level or below (Figure 5), which is enough for efficient cytokine binding to S100A6 which concentrations in serum reach the nanomolar level [68,70,71,72] (Appendix A). Moreover, the local levels of extracellular S100A6 in damaged tissues expressing S100A6 should be even higher, further facilitating S100A6–cytokine interactions.

Overall, S100A6 interacts with ca. 73% of the four-helical cytokines studied to date (32 of the 44 cytokines, see Table 1, Table 2 and Appendix A). The selectivity of S100A6 binding to the cytokines is equivalent to that of S100P [47], except for the interactions with IL-2 and OSM, which are specific only to S100A6 and S100P, respectively. Importantly, the revealed S100A6–cytokine interactions (Table 2) are non-redundant, as the S100A6-specific cytokines are mostly evolutionary distant from each other; the pairwise sequence identities within each of their SCOP families, calculated using Clustal Omega 2.1 (implemented in the EMBL-EBI service [12]), ranges from 3% to 44% (the exception is the GH–GH-V pair, with a pairwise sequence identity of 93%). According to the IntAct [73] and BioGRID [74] databases, so far, only the S100P protein has been described as a soluble non-receptor extracellular target protein for the following cytokines specific to S100A6 (Table 2): IL-3, IL-5, IL-9, IL-13, IL-21, THPO, and IL-22. Except for IL-22, they are short-chain four-helical cytokines, with the highest average affinity to S100A6 (Figure 5).

### 3.2. Structural Modeling of The S100A6–Cytokine Complexes

The docking models of the complexes between the Ca^2+^-loaded S100A6 dimer and the S100A6-specific four-helical cytokines shown in Table 1 and Table 2 (excluding the heterodimeric IL-27/IL-35) were generated using the ClusPro docking server [59], and analyzed as previously described [43,47]. The residues predicted to be involved in the interaction for five or more of the ten docking models are listed in Appendix A.

The S100A6 residues predicted to be key in the recognition of the four-helical cytokines (for at least half of the cytokines) are I44 (16 cytokines out of 30) of the ‘hinge’ loop region between helices α2 and α3, R55 and D59 of helix α3 (17 cytokines), and I83 (21 cytokines) and Y84 (18 cytokines) of helix α4 (Figure 6A). An analysis of the distribution of the predicted contact residues of the S100A6 dimer along its amino acid sequence within the models of its complexes with the four-helical cytokines (Figure 6B) shows that cytokines of the different structural families share the contact surfaces of S100A6, including *N*- and *C*-termini, helix α1, the ‘hinge’ region, and helices α3 and α4. These regions of the S100A6 dimer form a well-defined cytokine-binding site located between its subunits (Figure 6A), which is remarkably similar to that previously predicted for S100P interaction with the four-helical cytokines [47]. The qualitative difference between the predicted behavior of the S100A6 and S100P proteins is the more complete involvement in cytokine recognition of the helix α3 and the *N*-terminal half of helix α4 of S100A6. In this sense, the predicted pattern in the contact residues of S100A6 is more reminiscent of that described in S100 proteins, with frequent involvement in the target recognition of helix α1, the ‘hinge’, and helices α3 and α4 [75]. Importantly, only 4 of the 11 residues of S100A6 previously shown to interact with the V domain of RAGE using NMR [67] are predicted to interact with some of the cytokines (4 out of 30): R62 (GH-V, IL-11, IFN-β, IL-20), N63 (IL-11, IFN-β), K64 (IFN-β), and N69 (IL-11) (Appendix A). The RAGE-binding site of S100A6 reported in [76] is even more distinct from the predicted cytokine-specific site. Thus, the latter is notably different from the RAGE-binding site.

The molecular docking analysis indicates that the long-chain cytokines preferentially bind the Ca^2+^-bound S100A6 dimer via the residues of the *N*-terminus and helices α1 and α3 (Figure 7B). The cytokines of the interferon/IL-10 family are predicted to bind the S100A6 dimer via the residues of the *C*-terminus and helices α2 and α3 (Figure 7C). However, the predicted locations of the contact residues for the short-chain cytokines do not reveal clear regularities (Figure 7A). 

Furthermore, unlike other families of cytokines, this family tends to interact with only one chain of the S100A6 dimer. Therefore, similarly to the S100P binding to the four-helical cytokines [47], the putative location of the S100A6-binding site in the cytokines varies, depending on the structural family of the cytokine, and the particular cytokine. At the same time, analysis of the contact residues (Appendix A) indicates that arginines are most vastly involved in S100A6 binding, regardless of the cytokine family (14–22% of the contact residues).

The cytokine residues predicted to interact with S100A6 for some of the short-chain cytokines (GM-CSF, IL-3, IL-13, IL-21, THPO), long-chain cytokines (GH, GH-V, IL-31, PRL), and cytokines of the interferon/IL-10 family (IFN-ω1, IL-10, IL-20, IL-22, and IL-24) overlap by more than 50% with the residues similarly predicted for interaction with the S100P protein [47]. On the contrary, the putative S100A6-binding sites of specific short-chain cytokines (EPO, IL-5, IL-15), long-chain cytokines (LEP), and cytokines of the interferon/IL-10 family (IFN-β, IL-26) do not intersect with the sites predicted for S100P binding [43,47]. Thus, despite the nearly identical selectivity of the S100A6 and S100P proteins with regard to the binding of the four-helical cytokines, and the similar putative locations of their cytokine-binding sites, they could be recognized by the cytokines quite differently. 

As some of the predicted S100A6-binding residues of the specific four-helical cytokines (EPO, GM-CSF, IL-2, IL-3, IL-5, IL-13, IL-15, IL-21, GH, IFN-ω1, IL-20, IL-22, and IL-24) have previously been shown to participate in the recognition of their respective receptors (Appendix A), S100A6 binding is potentially able to affect the cytokine signaling, as demonstrated via the inhibition of IFN-β signaling in MCF-7 cells by S100A1/A4/B/P [25,44,45], the suppression of the cytotoxic activity of soluble TNF against Huh-7 cells by S100A12/A13 [42], the inhibition of the FGF2-induced increase in the proliferation of MCF-7/MDA-MB468 cells by S100B [40], the FGF2-medited activation of FGFR1 by S100B in myoblasts [41,48], and the S100A4 stimulation of the amphiregulin-mediated proliferation of embryonic fibroblasts [37].

Of note, the presented structural predictions based on the molecular docking suffer from their inability to take into consideration the structural flexibility inherent to the four-helical cytokines, as evidenced by the experimentally confirmed disorder in IL-15, THPO, G-CSF, LEP, and IL-10, and the theoretical predictions described in [47].

### 3.3. Potential Physiological Significance of the S100A6 Interactions with the Four-Helical Cytokines

To get an insight into the possible relative in vivo concentrations of S100A6 and the S100A6-specific cytokines (Table 1 and Table 2), we collected data from the literature on their concentrations in physiological fluids under normal and pathological conditions (Appendix A). An examination of the concentration ranges reported for S100A6 and the S100A6-specific cytokines in the blood serum/plasma (Figure 8) highlights several regularities. Firstly, there is an overlap in the concentration ranges for both interaction partners, which indicates the possibility of the mutual regulation of the activity of one of the partners due to an excess of the other partner. The binding of S100A6 could alter the cytokine signaling, as previously shown for IFN-β signaling in MCF-7 cells inhibited by S100A1/A4/B/P [25,44,45], and in other cases [37,40,42]. Alternatively, an excess of the S100A6-specific cytokine over the S100A6 protein could affect its interaction with RAGE [66], integrin β1 [29], and/or other receptors. Secondly, with a few exceptions, there is a trend towards increased cytokine concentrations in pathological conditions, which should promote the formation of the S100A6–cytokine complex. The most favorable conditions for that are expected for GH, GH-V, LEP, PRL, and IL-26, which demonstrate the highest blood levels, exceeding 1 nM (Figure 8). As mentioned above, although the experimental estimates of the equilibrium dissociation constants (Table 2 and Figure 5), K_d_, are low enough for efficient S100A6 binding only for THPO, S100A6 monomerization could decrease the K_d_ values by 1.4–2.2 orders of magnitude, which was earlier shown for the S100P interaction with IL-11 and IFN-β [25,64,69]. In this case, many other cytokines could bind S100A6 in vivo, at concentrations as high as several nM [70] (Appendix A).

The binding of S100A6 to the cytokines may also promote their non-canonical secretion, as previously reported for S100A13 binding to IL-1α and FGF1 [38,39] and for CLCF1, which requires CRLF1 binding for efficient secretion [77]. Considering that the sharing of interaction partners is inherent to many S100 proteins, including S100A6 and S100P [50,51], the revealed multiple interactions of the four-helical cytokines with S100A6/S100P, and other members of the S100 family [25,43,44,45,46,47,63,64], could serve as a common mechanism for the non-canonical secretion of the cytokines.

## 4. Conclusions

In this work, we have extended the list of the four-helical cytokines studied, with regard to their affinity to the promiscuous S100A6 protein [50,51], from nine (see references to Table 1) to forty-four (see also Appendix A) cytokines, covering all structural families of this fold. Only 12 out of the 44 cytokines studied using SPR spectroscopy lack a notable specificity to S100A6 (Appendix A). The absence of detectable interactions implies that the respective equilibrium dissociation constants, K_d_, exceed 10^−4^ M. Thus, about 73% of the cytokines are specific to the Ca^2+^-loaded S100A6 dimer, with K_d_ values in the range from 0.3 nM to 12 µM (Table 2), which intersect with the K_d_ estimates for the complexes of Ca^2+^-bound S100A6 with extracellular RAGE fragments from 28 nM to 13 µM [65,66,67]. The fraction of the cytokines specific to S100A6 may be even higher, given the possibility that S100A6 monomerization promotes its interaction with the cytokines. The similarly high percentage of the four-helical cytokines specific to the S100P protein of 71% [47] cannot be attributed to the high homology between S100A6 and S100P, as their pairwise sequence identity is only 35%.

The promiscuous binding of S100A6 and S100P to a wide range of the four-helical cytokines belonging to all structural families of this fold can be partly explained via the molecular docking analysis, which revealed that both S100 proteins have a nearly identical cytokine-binding site formed by helices α1, α3 and α4, and the ‘hinge’ (Figure 6A, [47]). The involvement of the ‘hinge’ and helix α4 in the cytokine recognition was previously confirmed via S100P mutagenesis [47]. This site overlaps with the RAGE-binding site of S100P, but differs markedly from that of S100A6. Meanwhile, the structural modeling shows that the S100-specific four-helical cytokines lack a conserved S100-binding site (Figure 7, [47]). Instead, the location of the putative S100A6/S100P-binding sites of the cytokines is variable, and depends on the particular cytokine. Importantly, the modelling reveals distinct differences in the cytokine regions putatively involved in the recognition of the S100A6 or S100P proteins, which explains some differences in their selectivity to the cytokines (limited to the recognition of IL-2 and OSM). Nevertheless, similarly to the putative S100P-binding sites of the four-helical cytokines [47], some of their predicted S100A6-specific regions are involved in binding of the corresponding receptors. This fact raises the possibility that the interaction of some cytokines with extracellular S100A6/S100P may interfere with the formation of the cytokine-receptor complexes, thereby inhibiting proper signaling (see, for instance, [25,40,42,44,45]). The same effect is expected for other promiscuous members of the S100 family, capable of interaction with multiple common binding partners [50,51]. In this case, the promiscuous S100 proteins are potentially able to serve as universal inhibitors of the signaling of the four-helical cytokines. This unique feature of the S100 family could be used to reduce the severity of the disorders associated with an excessive release of the cytokines, including cytokine storm. Another possibility is the S100-binding induced activation of cytokine signaling, as demonstrated via the FGF2-medited stimulation of FGFR1 by S100B in myoblasts [41,48], and the enhancement of the amphiregulin-induced proliferation of embryonic fibroblasts by S100A4 [37]. Moreover, the cytokine binding could modulate the extracellular activity of the promiscuous S100 proteins in response to the pathological conditions that are accompanied by elevated levels of some four-helical cytokines. Finally, the binding of intracellular S100 proteins could facilitate the non-canonical secretion of certain cytokines [38,39,77].

## Figures and Tables

**Figure 1 biomolecules-13-01345-f001:**
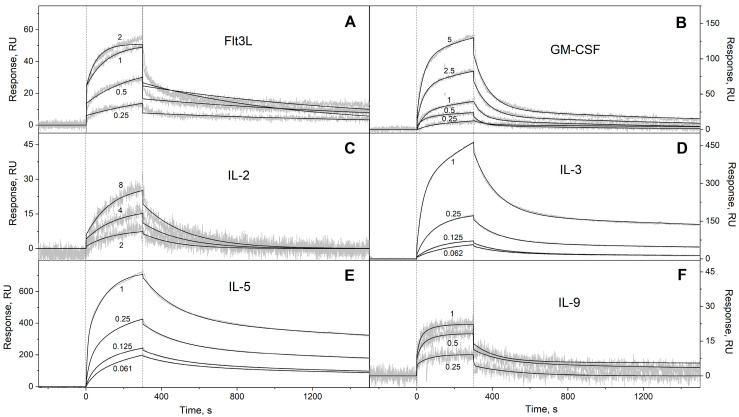
SPR spectroscopy data on the kinetics of association/dissociation of the complexes of Ca^2+^-bound S100A6 with the short-chain four-helical cytokines (see Table 2) immobilized on the sensor chip surface via amine coupling, at 25 °C: (**A**) Flt3L, (**B**) GM-CSF, (**C**) IL-2, (**D**) IL-3, (**E**) IL-5, (**F**) IL-9. The association phase is marked by the vertical dotted lines. The micromolar concentrations of S100A6 are indicated for the sensograms. The experimental curves (grey) are described via the heterogeneous ligand model (1) or the one-site binding model (black curves) (see Table 2).

**Figure 2 biomolecules-13-01345-f002:**
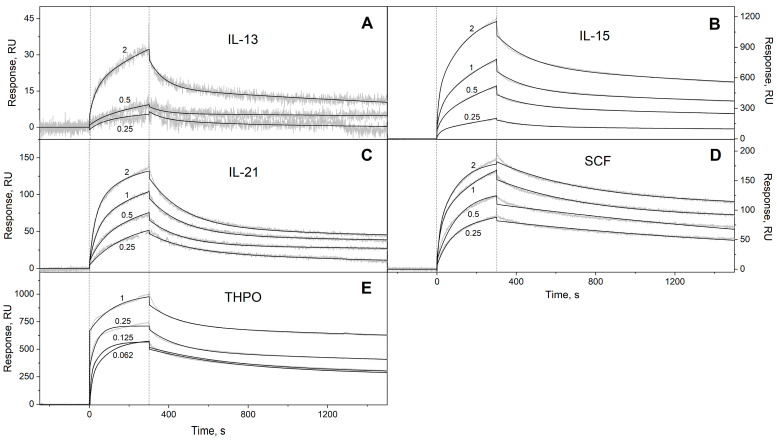
SPR spectroscopy data on the kinetics of association/dissociation of the complexes of Ca^2+^-bound S100A6 with the short-chain four-helical cytokines (see Table 2) immobilized on the sensor chip surface via amine coupling at 25 °C: (**A**) IL-13, (**B**) IL-15, (**C**) IL-21, (**D**) SCF, (**E**) THPO. The experimental curves (grey) are described via the heterogeneous ligand model (1) (black curves) (see Table 2). For other designations, refer to the caption to Figure 1.

**Figure 3 biomolecules-13-01345-f003:**
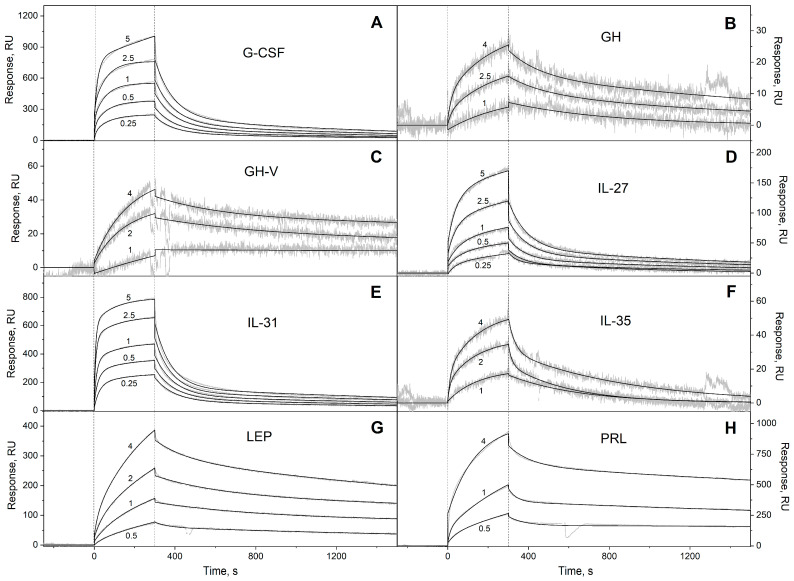
SPR spectroscopy data on the kinetics of association/dissociation of the complexes of Ca^2+^-loaded S100A6 with the long-chain four-helical cytokines (see Table 2) immobilized on the sensor chip surface via amine coupling, 25 °C: (**A**) G-CSF, (**B**) GH, (**C**) GH-V, (**D**) IL-27, (**E**) IL-31, (**F**) IL-35, (**G**) LEP, (**H**) PRL. The experimental curves (grey) are described via the heterogeneous ligand model (1) (black curves) (see Table 2). For other designations, refer to the caption to Figure 1.

**Figure 4 biomolecules-13-01345-f004:**
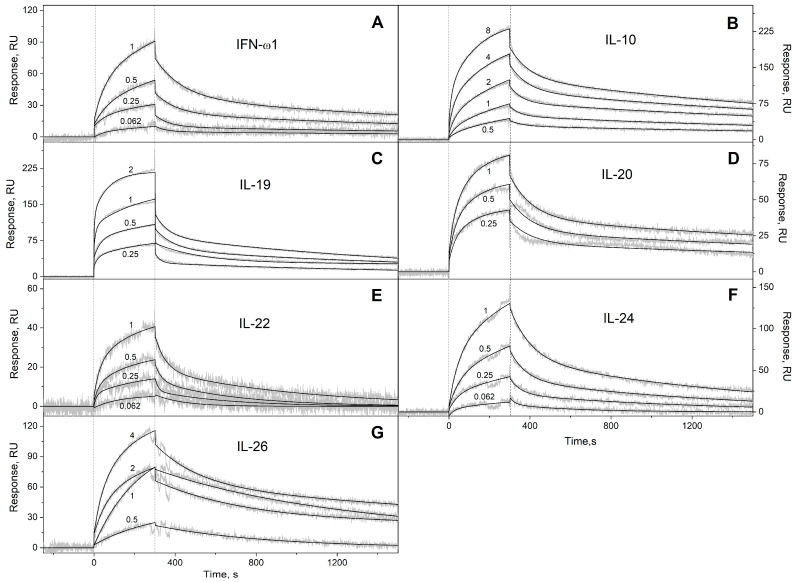
SPR spectroscopy data on the kinetics of association/dissociation of the complexes of Ca^2+^-bound S100A6 with the interferons/IL-10 four-helical cytokines (see Table 2) immobilized on the sensor chip surface via amine coupling at 25 °C: (**A**) IFN-ω1, (**B**) IL-10, (**C**) IL-19, (**D**) IL-20, (**E**) IL-22, (**F**) IL-24, (**G**) IL-26. The experimental curves (grey) are described via the heterogeneous ligand model (1) (black curves) (see Table 2). For other designations, refer to the caption to Figure 1.

**Figure 5 biomolecules-13-01345-f005:**
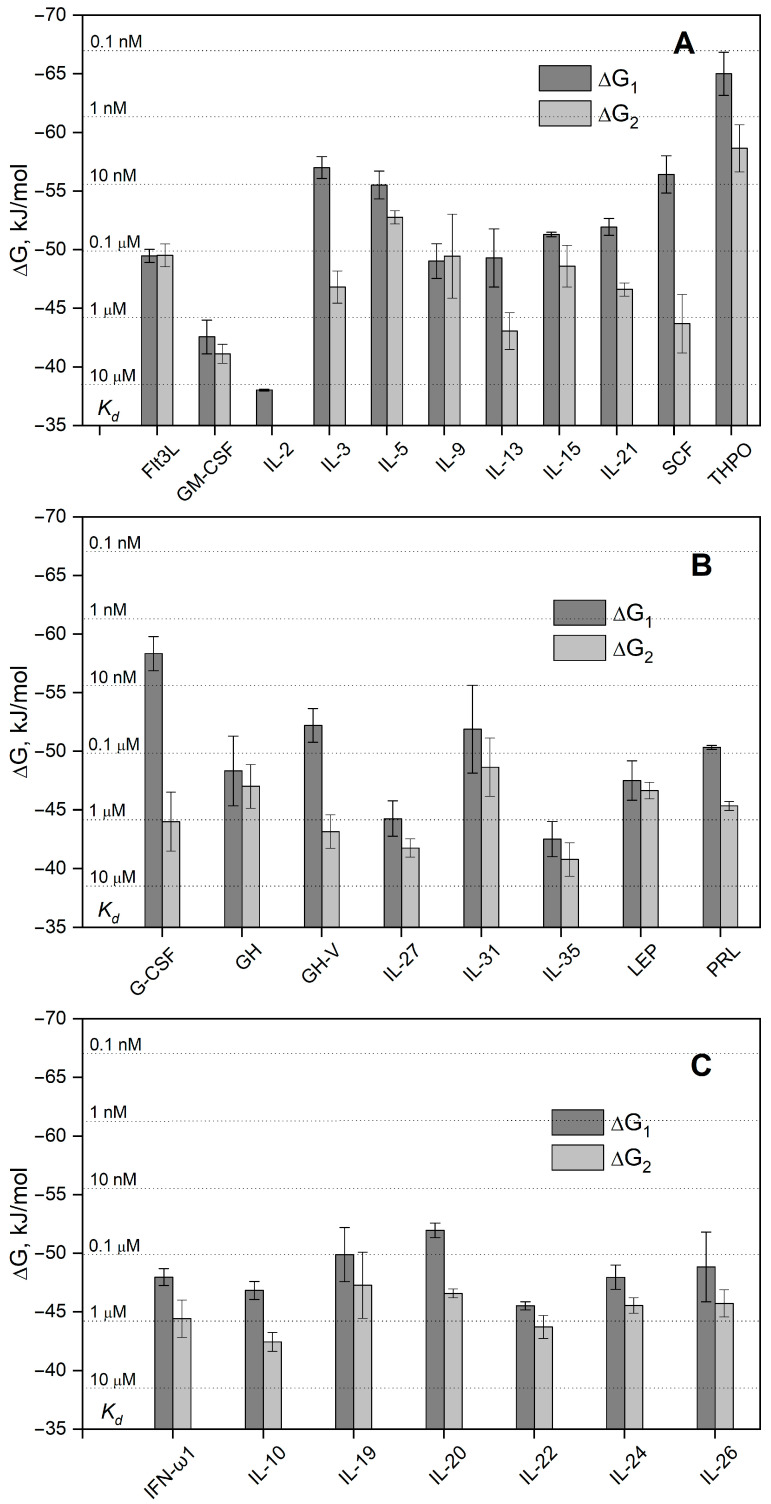
The free energy changes upon the binding of Ca^2+^-loaded S100A6 to the four-helical cytokines of short-chain (panel **A**), long-chain (**B**), or interferon/IL-10 (**C**) families at 25 °C, estimated from the SPR data shown in Table 2. The scale of K_d_ values is indicated.

**Figure 6 biomolecules-13-01345-f006:**
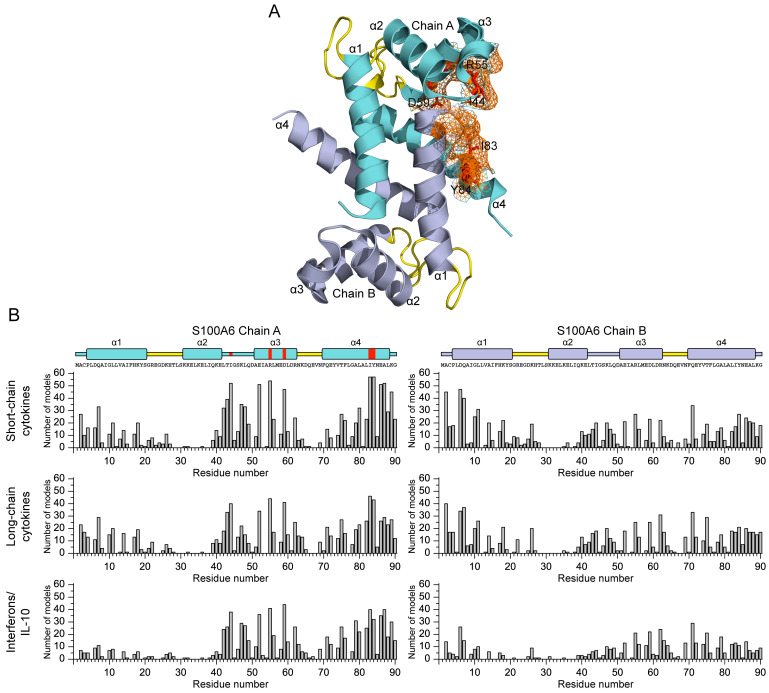
(**A**) The cytokine-binding site predicted for the Ca^2+^-loaded S100A6 dimer (PDB entry 1K9K) using the ClusPro docking server [59] and the tertiary structures listed in Appendix A. The chains A and B are highlighted in cyan and grey, respectively. The α-helices are labeled as α1–α4, and the Ca^2+^-binding loops are yellow-colored. The S100A6 residues predicted to recognize at least 10 out of the 30 four-helical cytokines (I44, E52, R55, D59, I83, Y84, E86, A87, and K89 of chain A, and A2 and D6 of chain B) are marked as an orange mesh surface, whereas the residues predicted to interact with at least half of the cytokines (I44, R55, D59, I83, and Y84) are shown as red balls and sticks. (**B**) Distributions of the predicted contact residues of S100A6 over its amino acid sequence, within the docking models of S100A6 complexes with the four-helical cytokines shown in Appendix A (10 models per each S100A6–cytokine pair). The boundaries of the α-helices α1–α4 are extracted from PDB entry 1K9K; the residues I44, R55, D59, I83, and Y84 of chain A are marked in red.

**Figure 7 biomolecules-13-01345-f007:**
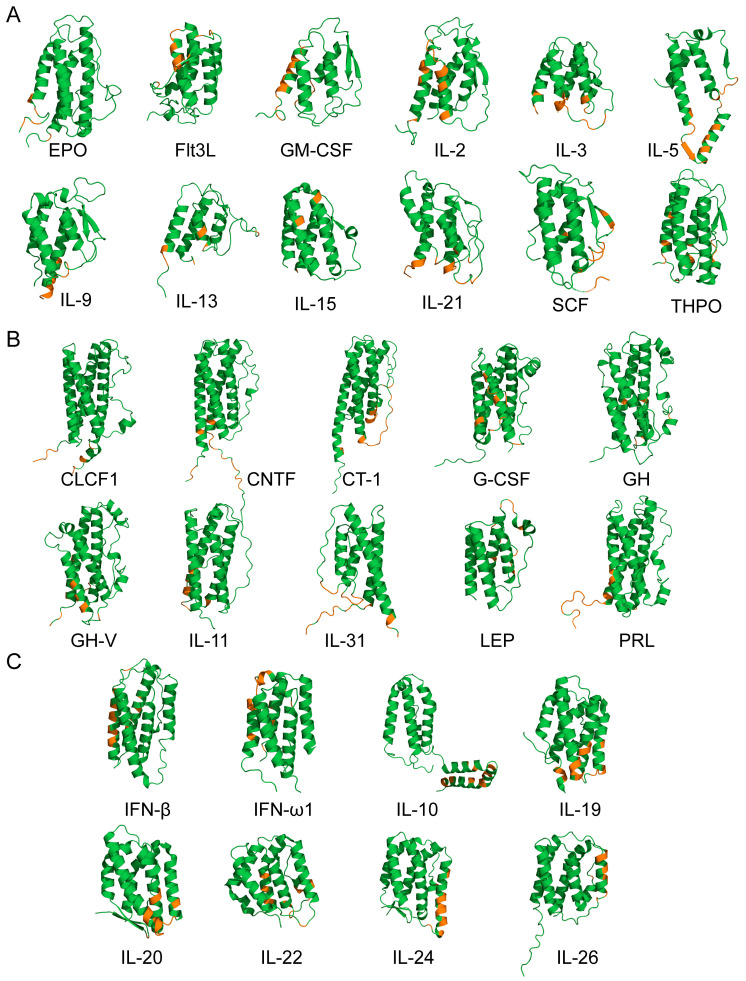
The S100A6-binding sites (orange-colored; see Appendix A) predicted for the four-helical cytokines of the short-chain (**A**), long-chain (**B**), and interferon/IL-10 (**C**) families, using the ClusPro docking server [59], and the tertiary structures listed in Appendix A. Only one subunit of IL-5 is shown. The *N*-terminus of each cytokine is located in the lower left corner. The analogous predictions for the S100P protein are presented in [47].

**Figure 8 biomolecules-13-01345-f008:**
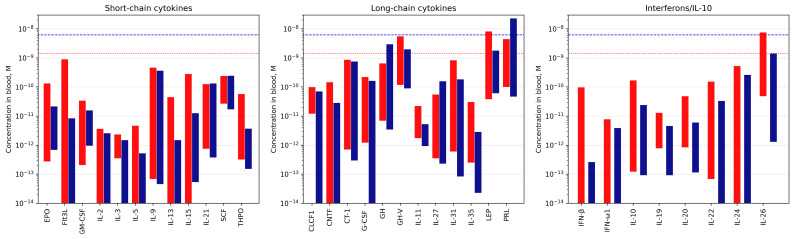
The concentration ranges of the S100A6-specific cytokines in the blood serum or plasma under normal (blue bars) and pathological (red bars) conditions, extracted from the literature data (see Appendix A). The blue and red dotted lines correspond to the upper limits of S100A6 serum levels in health and disease, respectively, according to the literature data (see Appendix A).

**Table 1 biomolecules-13-01345-t001:** The literature data on the equilibrium dissociation constants for the complexes between Ca^2+^-loaded S100A6 and four-helical cytokines at 25 °C. The SPR spectroscopy data (the amine coupling of the cytokine on the SPR chip surface) are described using the heterogeneous ligand model (1).

Cytokine	K_d1_, M	K_d2_, M	Reference
Short-chain cytokines			
EPO	(1.5 ± 0.3) × 10^−7^	(6.5 ± 2.6) × 10^−7^	[43]
Long-chain cytokines			
CLCF1	(1.1 ± 0.8) × 10^−6^	(3.7 ± 0.8) × 10^−6^	[46]
CNTF	(1.1 ± 1.0) × 10^−7^	(2.07 ± 0.08) × 10^−6^	[46]
CT-1	(1.6 ± 0.5) × 10^−6^	(1.2 ± 0.4) × 10^−5^	[46]
IL-11	(8.3 ± 1.9) × 10^−6^	(7.9 ± 2.1) × 10^−6^	[46]
Interferons/IL-10			
IFN-β	(8.2 ± 2.4) × 10^−8^ (0.70 ± 0.03) × 10^−9^ *	(2.67 ± 0.57) × 10^−7^ (2.81 ± 0.49) × 10^−7^ *	[44]

* S100A6 serves as a ligand.

**Table 2 biomolecules-13-01345-t002:** The parameters of the interactions between Ca^2+^-bound S100A6 and the specific four-helical cytokines shown in Appendix A at 25 °C, derived from the SPR spectroscopy data (Figure 1, Figure 2, Figure 3 and Figure 4), using the heterogeneous ligand model (1).

Cytokine	k_d1_, s^−1^	K_d1_, M	k_d2_, s^−1^	K_d2_, M
Short-chain cytokines				
Flt3L	(8.83 ± 2.54) × 10^−4^	(1.22 ± 0.27) × 10^−7^	(8.32 ± 4.05) × 10^−4^	(1.25 ± 0.46) × 10^−7^
GM-CSF	(7.92 ± 4.02) × 10^−4^	(2.26 ± 1.18) × 10^−6^	(1.76 ± 0.49) × 10^−2^	(3.63 ± 1.14) × 10^−6^
IL-2 *	(4.97 ± 0.78) × 10^−3^	(1.20 ± 0.04) × 10^−5^	n/a	n/a
IL-3	(2.71 ± 0.78) × 10^−4^	(6.11 ± 2.19) × 10^−9^	(7.28 ± 0.65) × 10^−3^	(4.02 ± 2.03) × 10^−7^
IL-5	(2.47 ± 0.45) × 10^−4^	(1.16 ± 0.52) × 10^−8^	(5.07 ± 0.20) × 10^−3^	(3.23 ± 0.73) × 10^−8^
IL-9	(5.97 ± 2.79) × 10^−3^	(1.69 ± 0.90) × 10^−7^	(5.85 ± 2.72) × 10^−3^	(2.71 ± 2.42) × 10^−7^
IL-13	(2.64 ± 0.98) × 10^−4^	(1.97 ± 1.50) × 10^−7^	(1.36 ± 0.06) × 10^−2^	(1.91 ± 1.07) × 10^−6^
IL-15	(1.62 ± 0.24) × 10^−4^	(5.70 ± 0.46) × 10^−8^	(6.29 ± 0.26) × 10^−3^	(2.15 ± 1.32) × 10^−7^
IL-21	(1.88 ± 1.33) × 10^−4^	(4.57 ± 1.30) × 10^−8^	(5.68 ± 0.34) × 10^−3^	(3.87 ± 0.86) × 10^−7^
SCF	(2.57 ± 0.92) × 10^−4^	(8.72 ± 4.95) × 10^−9^	(4.51 ± 1.52) × 10^−3^	(1.90 ± 1.45) × 10^−6^
THPO	(6.00 ± 3.29) × 10^−5^	(2.89 ± 1.83) × 10^−10^	(4.47 ± 2.46) × 10^−3^	(3.96 ± 2.65) × 10^−9^
Long-chain cytokines				
G-CSF	(7.72 ± 0.31) × 10^−4^	(3.91 ± 2.06) × 10^−9^	(1.04 ± 0.04) × 10^−2^	(1.68 ± 1.29) × 10^−6^
GH	(3.73 ± 2.36) × 10^−4^	(3.42 ± 2.85) × 10^−7^	(5.87 ± 1.16) × 10^−3^	(4.16 ± 2.65) × 10^−7^
GH-V	(7.92 ± 3.35) × 10^−5^	(4.62 ± 2.41) × 10^−8^	(3.06 ± 1.27) × 10^−3^	(1.78 ± 0.92) × 10^−6^
IL-27 ^#^	(9.61 ± 4.18) × 10^−4^	(1.16 ± 0.63) × 10^−6^	(1.47 ± 0.62) × 10^−2^	(2.80 ± 0.85) × 10^−6^
IL-31	(4.64 ± 0.13) × 10^−4^	(1.06 ± 0.96) × 10^−7^	(1.11 ± 0.12) × 10^−2^	(2.56 ± 1.95) × 10^−7^
IL-35 ^#^	(2.59 ± 0.81) × 10^−3^	(2.34 ± 1.26) × 10^−6^	(3.91 ± 1.55) × 10^−2^	(4.65 ± 2.42) × 10^−6^
LEP	(3.18 ± 2.17) × 10^−3^	(3.25 ± 1.92) × 10^−7^	(2.87 ± 1.00) × 10^−4^	(3.85 ± 1.06) × 10^−7^
PRL	(1.59 ± 0.23) × 10^−4^	(8.40 ± 0.58) × 10^−8^	(1.55 ± 0.82) × 10^−2^	(6.35 ± 0.96) × 10^−7^
Interferons/IL-10				
IFN-ω1	(4.71 ± 0.22) × 10^−4^	(2.28 ± 0.64) × 10^−7^	(1.29 ± 0.55) × 10^−2^	(1.11 ± 0.63) × 10^−6^
IL-10	(3.98 ± 0.28) × 10^−4^	(3.61 ± 1.09) × 10^−7^	(1.50 ± 0.37) × 10^−2^	(2.13 ± 0.67) × 10^−6^
IL-19	(6.28 ± 0.86) × 10^−4^	(1.47 ± 1.08) × 10^−7^	(2.77 ± 1.69) × 10^−2^	(4.96 ± 4.04) × 10^−7^
IL-20	(3.08 ± 0.86) × 10^−4^	(4.51 ± 1.12) × 10^−8^	(9.14 ± 2.51) × 10^−3^	(3.85 ± 0.59) × 10^−7^
IL-22	(3.57 ± 2.56) × 10^−3^	(5.93 ± 0.84) × 10^−7^	(3.26 ± 2.67) × 10^−2^	(1.31 ± 0.49) × 10^−6^
IL-24	(5.73 ± 0.85) × 10^−4^	(2.39 ± 0.94) × 10^−7^	(7.79 ± 1.72) × 10^−3^	(6.01 ± 1.57) × 10^−7^
IL-26	(5.98 ± 2.92) × 10^−4^	(2.78 ± 2.31) × 10^−7^	(2.56 ± 1.39) × 10^−3^	(6.00 ± 2.63) × 10^−7^

* a single-site binding model is used; #, heterodimeric cytokines; n/a, not applicable.

## Data Availability

The data present in the current study are available from the corresponding authors on reasonable request.

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
