# Peer review of "Interaction of S100A6 Protein with the Four-Helical Cytokines"

_biomolecules, 2023, doi:10.3390/biom13091345_

Round 1

Reviewer 1 Report

In this manuscript, Kazakov et al studied by SPR the interactions of S100A6 with various cytokines. The research design is similar to their previous work about S100P interactions with cytokines. However the authors also reviewed the literature for extracting the concentrations of cytokines under normal and pathological conditions. This section highlights the potential role of S100A6 for cytokine buffering or signaling. The research design and the results are clearly presented and therefore I have few minor remarks.

1. From Table S4, what are the characteristics of the cytokines regions recognized by S100A6. Are they amphipathic helices, is there some consensus motifs?

2. In the binding model of cytokines on S100A6, if S100A6 is dimeric, is there some cooperative effect of cytokine binding?

Author Response

In this manuscript, Kazakov et al studied by SPR the interactions of S100A6 with various cytokines. The research design is similar to their previous work about S100P interactions with cytokines. However the authors also reviewed the literature for extracting the concentrations of cytokines under normal and pathological conditions. This section highlights the potential role of S100A6 for cytokine buffering or signaling. The research design and the results are clearly presented and therefore I have few minor remarks.

  1. From Table S4, what are the characteristics of the cytokines regions recognized by S100A6. Are they amphipathic helices, is there some consensus motifs?

ANSWER: Unfortunately, we were unable to reveal vivid structural regularities of the predicted S100A6-binding sites. Meanwhile, analysis of the contact residues indicates that arginines are most massively involved in the S100A6 binding, regardless of the cytokine family (14-22% of the contact residues). This phrase has been added to chapter 3.2 “Structural modeling of the S100A6-cytokine complexes”.

  1. In the binding model of cytokines on S100A6, if S100A6 is dimeric, is there some cooperative effect of cytokine binding?

ANSWER: Our attempts to use the scheme of bivalent analyte for description of the SPR data in many cases showed a considerably worse quality of the fits. In fact, this binding model suggests structural integrity of S100A6 dimer and complete independence of its cytokine-binding sites. The both conditions are unlikely to be met in this particular case.

Reviewer 2 Report

The authors show that S100A6 interacts with several  four-helical cytokines , similar to other members of the S100 protein family. The manuscript presents no major flaws. However, different from other S100 cases, no functional correlates of this interaction is presented; only a certain amount of speculation. Also, the authors may wish to mention that S10B binds to FGFR1-bound FGF-2 in high-density myoblasts to promote proliferation and inhibit myogenic differentiation (PMID: 21693575; PMID: 22276098). Thus, the abstract sentence "This unique feature of the S100 proteins potentially allows them to serve as universal inhibitors of signaling of the four-helical cytokines " appears not to be supported by published results, at least as far as S100B is concerned.

Minor editing of English language required.

Author Response

The authors show that S100A6 interacts with several  four-helical cytokines , similar to other members of the S100 protein family. The manuscript presents no major flaws. However, different from other S100 cases, no functional correlates of this interaction is presented; only a certain amount of speculation.

ANSWER: We have identified so many S100A6-cytokine interactions (26) that it becomes highly problematic to cover their physiological significance in sufficient detail in a single paper. We consider this work as a basis for further experimental studies in this direction with participation of other research groups specializing in cellular studies.

Also, the authors may wish to mention that S10B binds to FGFR1-bound FGF-2 in high-density myoblasts to promote proliferation and inhibit myogenic differentiation (PMID: 21693575; PMID: 22276098). Thus, the abstract sentence "This unique feature of the S100 proteins potentially allows them to serve as universal inhibitors of signaling of the four-helical cytokines " appears not to be supported by published results, at least as far as S100B is concerned.

ANSWER: We have added these references (PMIDs 21693575 and 22276098) to the Introduction, section 3.2 “Structural modeling of the S100A6-cytokine complexes”, and Conclusions. We discussed the FGF2-mediated effects of S100B in the Conclusions chapter, and rephrased the final sentence of the abstract: “This unique feature of the S100 proteins potentially allows them to modulate activity of the numerous four-helical cytokines in the disorders accompanied by excessive release of the cytokines”.